# Bacteria producing the bioplastic polyhydroxybutyrate kill the nematode *Caenorhabditis elegans*

Gabrielle E. Giese[1]☉, Daniel M. Richards[1]☉, Jeremy T. Florman[2], Alyxandra N. Starbard[1], Ann A. Xu[1], Daniel J. Durning[3], Mark J. Alkema[2], Albertha J. M. Walhout[1]*

1 Department of Systems Biology, University of Massachusetts Chan Medical School, Worcester, Massachusetts, United States of America, 2 Department of Neurobiology, University of Massachusetts Chan Medical School, Worcester, Massachusetts, United States of America, 3 RNA Therapeutics Institute, University of Massachusetts Chan Medical School, Worcester, Massachusetts, United States of America

☉ These authors contributed equally to this work.
* marian.walhout@umassmed.edu

## Abstract

Bacteria, both individually and as symbionts of other organisms, significantly influence ecosystems by providing nutrients and metabolizing exogenous compounds. Some bacteria polymerize small organic acids such as lactate, pyruvate, and β-or 3-hydroxybutyrate when there is an excess of carbon relative to other elements. One such polymer, poly-β-hydroxybutyrate (PHB) is a biodegradable bioplastic. While the role of PHB as energy/carbon-storage in bacteria is well documented, the effects of PHB on interactions between bacteria and their hosts remain unclear. Here, we discover that PHB-producing bacteria can kill the nematode *Caenorhabditis elegans*. Death results from a combination of pharyngeal deformation, intestinal distention, disruption of the intestinal barrier, and defecation defects. Remarkably, mutations in *C. elegans nuc-1*, which encodes DNAse II, partially alleviate PHB-induced lethality. Altogether, our findings illustrate that PHB-producing bacteria can affect host-physiology and survival.

## Introduction

Bacterial metabolism plays an important role not only in the physiology of individual bacteria, but also in shaping bacterial communities across diverse environments, and in the physiology of human and animal hosts. When presented with a nutritional environment that is rich in carbon, but lacking other elements such as nitrogen, some bacteria can store the excess carbon in the form of energy-rich polymers. For instance, the small organic acids lactate, pyruvate, and the ketone body β-hydroxybutyrate can be converted and polymerized into the bioplastic poly-β-hydroxybutyrate (PHB) that is stored in large granules named carbonosomes [1]. PHB is a bioplastic that is often used for commercial applications [2]. While PHB has

**Data availability statement:** All relevant data are available within the manuscript and Supporting information files. Raw data can be found in S1 Data. Original code available on Github (https://github.com/JeremyFlorman/IntestinalCalcium) and Zenodo (https://zenodo.org/records/18942936).

**Funding:** This work was supported by the National Institutes of Health grants DK068429 (A.J.M.W., https://www.niddk.nih.gov/) and GM140480 (M.J.A., https://www.nigms.nih.gov/). *Caenorhabditis elegans* strains were obtained from the *C. elegans* Genome Center (CGC), which is funded by the NIH Office of Research Infrastructure Programs (P40 OD010440, https://orip.nih.gov/). The UMass Chan Medical School Electron Microscopy Core is funded by the National Center for Research Resources (S10 OD025113-01, https://grants.nih.gov/funding/find-a-fit-for-your-research/nih-institutes-centers-offices/OD). The funders had no role in study design, data collection and analysis, decision to publish, or preparation of the manuscript.

**Competing interests:** I have read the journal's policy and the authors of this manuscript have the following competing interests: MJA is a member of PLOS Biology's Editorial Board. The other authors declare no competing interests exist.

**Abbreviations:** 3-HB, 3-hydroxybutyrate; 3HP, 3-hydroxypropionate;Dar, deformed anal region; DMP, defecation motor program; EMS, ethyl methanesulfonate; EtBr, ethidium bromide; FPS, frames per second; GC-MS, gas chromatography-mass spectrometry; gDNA, genomic DNA; IldP, lactate permease; MIC, minimal inhibitory concentrations; NGM, nematode growth media; PCR, polymerase chain reaction; PFA, paraformaldehyde; PHB, poly-β-hydroxybutyrate; PTFE, polytetrafluoroethylene; PTS, phosphotransferase system; TCA, tricarboxylic acid; TEM, transmission electron microscopy.

been well characterized in the context of bacterial metabolism, little is known about if and how these polymers, which can make up as much as 50% of the dry cell weight of some bacteria, impact their eukaryotic hosts.

The nematode *Caenorhabditis elegans* is a bacterivore and different bacteria can be beneficial or detrimental to the animal, depending on nutritional or chemical context. For instance, when compared to animals fed the standard laboratory diet of *Escherichia coli*, animals fed the soil bacteria *Comamonas aquatica* DA1877 (hereafter referred to as *Comamonas*) develop faster, have fewer offspring, and shorter lifespans [3]. Furthermore, animals fed *Comamonas* are less sensitive to a buildup of propionic acid and 3-hydroxymethylbutyrate and the chemotherapeutic 5-fluoro-2′-deoxyuridine, but more sensitive to camptothecin and tamoxifen [4–8].

Here, we tested the effects of additional metabolites and bacteria on *C. elegans* physiology. We found that animals die when fed *Comamonas,* but not *E. coli*, supplemented with high concentrations of lactate or pyruvate. Using genetic screens in *Comamonas*, we found that the lethality induced by these metabolites is the result of bacterial PHB biosynthesis. *E. coli* does not have the PHB biosynthetic pathway, and by using an *E. coli* strain engineered to synthesize PHB, we show that PHB synthesis is sufficient to kill the animal. Impairment of at least three mechanisms contributes to eliciting death: physical PHB accumulation leading to obstruction of the pharynx, disruption of the intestinal barrier, and disruption of the defecation motor program (DMP). By performing a forward genetic screen in *C. elegans* we found that mutations in *nuc-1*, a DNAse II homolog, partially alleviate toxicity induced by PHB-producing bacteria.

## Results

### *Comamonas* kills *C. elegans* on high concentrations of lactate or pyruvate

To identify interactions among bacterial diet and metabolites we tested the effects of eight different metabolites from central carbon metabolism in combination with nine different bacteria on *C. elegans* growth and survival (S1 Table). Animals were fed different bacterial diets while growing on metabolite-supplemented culture medium and assessed for developmental timing relative to control animals fed the same diets but without supplemented metabolites. Each of the metabolites tested elicited toxicity in the animal, and toxicity was modulated by bacterial diet (Figs 1A and S1A). As expected, we observed less sensitivity to propionate or HMB in animals fed *Comamonas*, relative to those fed *E. coli* [4,8]. However, animals fed *Comamonas* were highly sensitive to toxicity resulting from lactate, pyruvate, and to a lesser extent, butyrate supplementation, while animals fed other bacteria were either not sensitive (lactate, pyruvate) or less sensitive (butyrate) (Figs 1A and S1A). Only the *Pseudomonas aeruginosa* PA14 mutant had a more general effect on overall metabolite toxicity, which, even though it is mutant for the virulence regulatory factor, *gacA,* may be due to residual pathogenicity [9].

Lactate and pyruvate are common metabolites generated by glycolysis, and we were surprised to find that these metabolites became so toxic to *C. elegans* when fed *Comamonas*. Indeed, when fed *Comamonas* supplemented with 75 mM lactate,

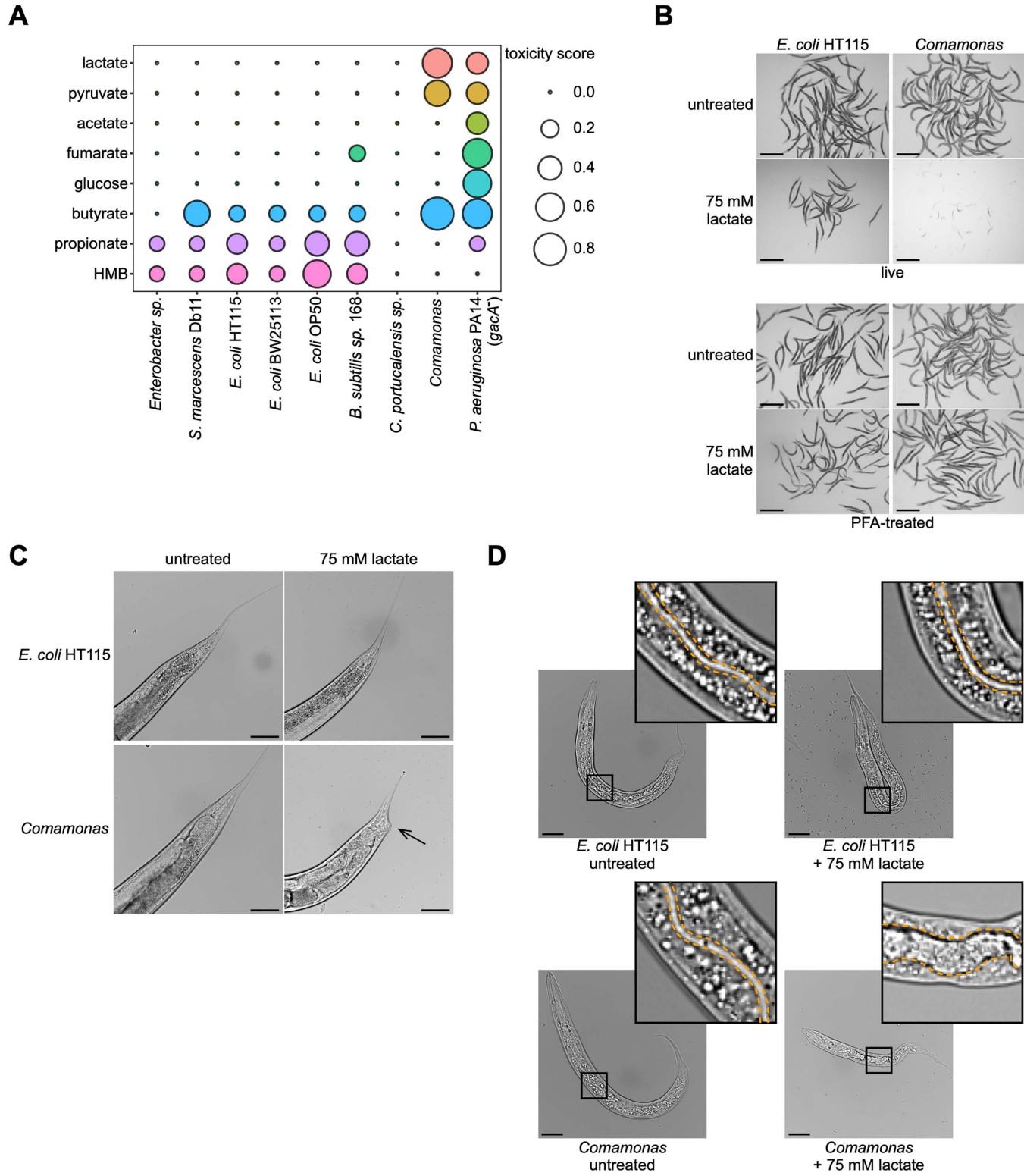

**Fig 1. Metabolite toxicity in *C. elegans* fed different bacterial diets. (A)** Bubble plot of metabolite toxicity scores of animals fed different bacteria. Higher score size indicates greater toxicity (see Methods). Minimal inhibitory concentrations (MICs) were determined qualitatively based on

developmental stage compared to the untreated control. HMB, 3-hydroxy-3-methylbutyrate. Data underlying this plot can be found in S1 Data. **(B)** Bright-field images of animals taken after 72 hours after feeding. Scale bar, 500 µm. PFA, paraformaldehyde. **(C)** Brightfield images of adult animals. Arrow indicates the deformed anal region (Dar) phenotype. Scale bar, 50 µm. **(D)** Brightfield images of animals. Orange dashed lines depict intestinal lumen. Scale bar, 25 µm.

most first larval (L1) stage animals died or arrested development (Fig 1B, top). Animals that did survive the lactate supplemented *Comamonas* diet developed a deformed anal region (Dar) phenotype (Fig 1C), which has been previously observed in *C. elegans* fed the nematode-specific pathogen *Microbacterium nematophilum* [10]. Both arrested L1 and the occasional surviving older animals showed a distortion of the pharynx and of the distention intestinal lumen (Figs 1D and S1B). We fed animals paraformaldehyde (PFA) treated, metabolically inactive *Comamonas* (see Methods) supplemented with lactate, and found that this alleviated lactate-induced toxicity. This result indicates that active bacterial metabolism is required to convert lactate into a toxic compound (Fig 1B-bottom).

### The *Comamonas* PHB biosynthesis pathway is required for lactate toxicity in *C. elegans*

To understand how bacterial metabolism kills *C. elegans* supplemented with lactate we screened a collection of ~5,500 *Comamonas* transposon insertion mutant strains [11]. We identified 56 *Comamonas* mutants that (partially) suppressed lactate-induced animal death (Fig 2A; S2 Table). Visual inspection of the mutated genes revealed six genes that act directly in the PHB biosynthesis pathway and seven others that function either in connected pathways or in the regulation of PHB synthesis (Fig 2B). PHB is a biopolymer comprised of the monomer β- or 3-hydroxybutyrate (3-HB), produced by many bacterial species as an intracellular carbon-storage molecule. PHB is generated when carbon sources such as lactate or pyruvate are high but some other element (*e.g.,* nitrogen) is deficient [12,13]. We found that 48 of the *Comamonas* mutants (86%) also suppressed pyruvate toxicity, which could be expected since these metabolites can be interconverted by lactate dehydrogenase (Fig 2B, 2C, and S2 Table). Notably, a mutation in lactate permease (hereafter referred to as *lldP*), annotated as a lactate transporter, also rendered animals resistant to pyruvate, suggesting that this transporter can also transport pyruvate. We also tested direct supplementation of the 3-HB monomer and found that it was toxic as well, including in animals fed *lldP* mutant bacteria, indicating that this metabolite is not taken up via this transporter. While *Comamonas aceE* mutants rescued lactate and pyruvate toxicity, 3-HB toxicity was retained in animals fed this mutant (Fig 2C). *aceE* encodes the E1 component of pyruvate dehydrogenase, which converts pyruvate into acetyl-CoA in the first step of the PHB biosynthetic pathway (Fig 2B). These observations are consistent with the annotation that both *lldP* and *aceE* act upstream of the reaction that polymerizes 3-HB (Fig 2B). 3-HB toxicity was prevented, however, by feeding animals a *Comamonas phaC* mutant, which encodes the enzyme that polymerizes 3-HB into PHB (Fig 2B). Our screen also uncovered the *Comamonas phaP* gene, which encodes phasin, an important protein component of PHB required for granule formation [14], and mutations in *phaP* conferred resistance to all three metabolites (Fig 2C). Finally, *E. coli* HT115, which does not contain the genes for PHB production, causes neither *C. elegans* lethality nor the Dar phenotype when supplemented with lactate (Figs 2C and 1C).

When L1 animals were fed wild-type *Comamonas* supplemented with lactate, 73% arrested, 19% were dead, and 8% survived, while animals fed *Comamonas phaC* mutants all survived (Fig 2D). After starvation or exposure to certain pathogens arrested larvae can continue to develop when they are fed other, nontoxic bacteria, or when they are removed from the pathogen [15,16]. Therefore, we tested whether development would proceed when animals were transferred from PHB-producing *Comamonas* to non-PHB-producing *Comamonas*. While ~25% of animals developed after transfer to non-PHB-producing *Comamonas* following a 24-hour exposure to PHB-producing bacteria, almost no animals survived after 72-hour exposure (Fig 2E). As a control, nearly 100% of animals that were starved and then transferred to non-PHB-producing *Comamonas* continued to develop even after 72 hours. These results indicate that PHB-producing bacteria arrest *C. elegans* development, prevent recovery, and ultimately kill the animal.

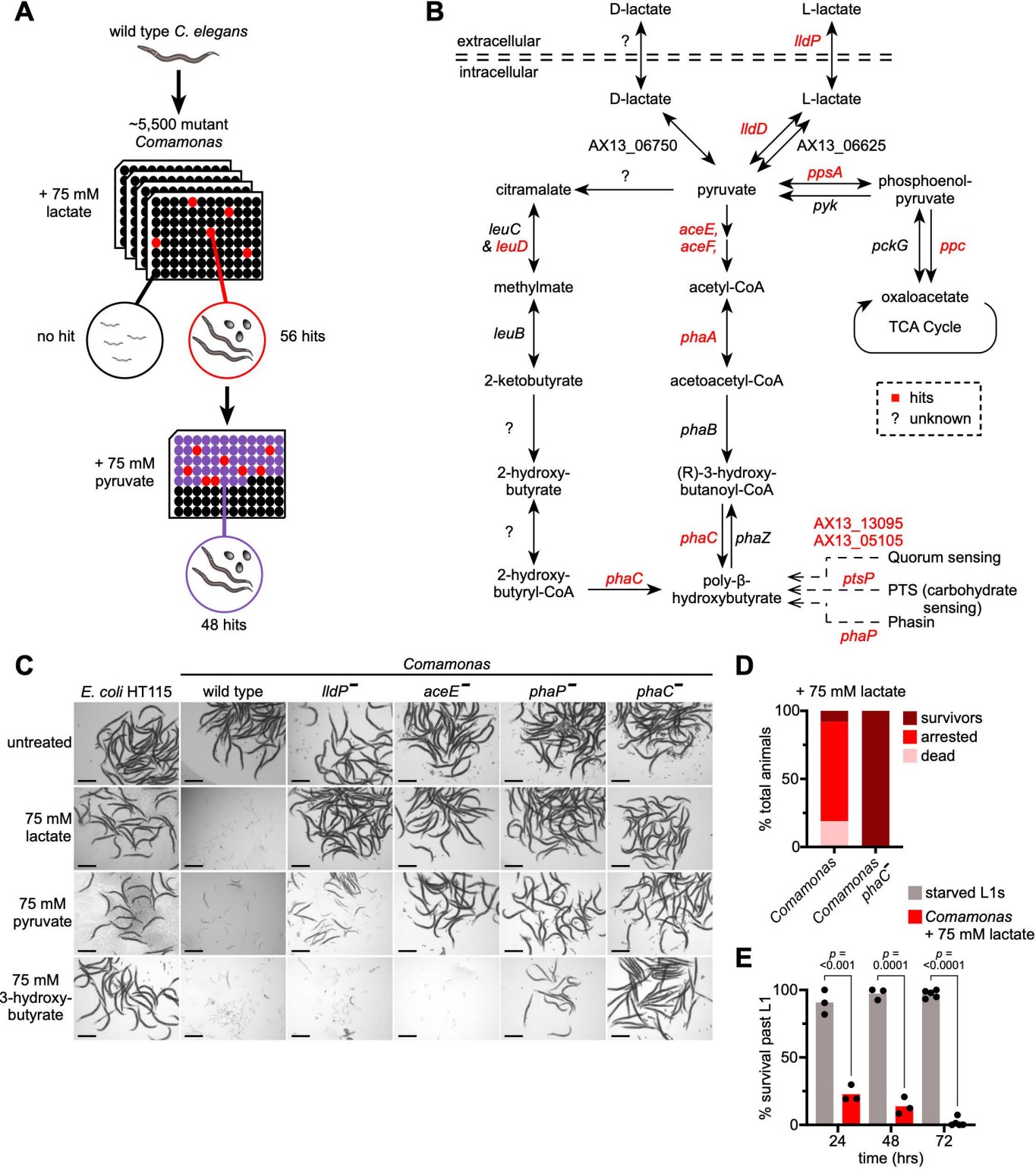

**Fig 2. The *Comamonas* PHB biosynthesis pathway is required to elicit lactate toxicity in *C. elegans*. (A)** Schematic of *Comamonas* mutant screen. **(B)** *Comamonas* PHB biosynthesis pathway. TCA, tricarboxylic acid; PTS, phosphotransferase system. **(C)** Brightfield images of *C. elegans*

grown on *E. coli* HT115, *Comamonas,* or select *Comamonas* mutants with different PHB-precursors for 72 hours. Scale bar, 50 μm. **(D)** Quantification of *C. elegans* phenotypes on *Comamonas* wild-type or a PHB-deficient *Comamonas* mutant supplemented with lactate. **(E)** Survival of animals transferred to *Comamonas* without lactate after being cultured on *Comamonas* supplemented with lactate for 24, 48, or 72 hours. *p* values determined by unpaired Welch's *t* test. The data underlying this figure can be found in S1 Data.

## Bacterial PHB production is sufficient to kill *C. elegans* in different bacterial contexts

*Cupriavidus necator* H16 bacteria (formerly known as *Ralstonia eutropha* H16) produce high levels of PHB and have been extensively used in both basic research and industrial PHB production [17,18]. Similar to a *Comamonas* plus lactate diet, the majority of L1 animals failed to grow when fed *C. necator* supplemented with lactate, and the few survivors developed a Dar phenotype (Figs 3A and S2A). In contrast, when fed *C. necator* without metabolite supplementation, *C. elegans* development was only slightly delayed, and this delay was variable among individuals, similar to the effect of feeding *Comamonas* supplemented with a lower dose of lactate (Figs 3A, S2B, and S2C). This indicates that PHB-producing bacteria are toxic to *C. elegans*.

PHB forms carbonosomes that can be observed as large granules by transmission electron microscopy (TEM) [19]. Using TEM, we observed granules in both lactate and untreated conditions in *C. necator,* and in wild-type *Comamonas*, but not *Comamonas lldP* mutants or *E. coli* supplemented with lactate (Fig 3B). To measure PHB more directly, we used gas chromatography coupled to mass spectrometry (GC-MS). As expected, *C. necator* produced a high amount of PHB when supplemented with lactate (Fig 3C). Some PHB was also detected in *C. necator* without lactate supplementation, and although the PHB levels were similar to those in *Comamonas* supplemented with lactate, *C. necator* without lactate supplementation was less toxic to *C. elegans,* indicating that *Comamonas* confers additional, PHB-independent toxicity in response to lactate supplementation (Figs 3A, 3C and S2A). During the course of our experiments, we observed some heterogeneity in the amount of PHB produced by *C. necator.* Therefore, we measured PHB from five independent *C. necator* colonies and found that the colony that produced the highest level of PHB also elicited the strongest developmental delay phenotype in *C. elegans* (S2D Fig). Importantly, GC-MS measurements showed that PHB was significantly induced by lactate in *Comamonas* and could not be detected in *E. coli* HT115 with or without lactate supplementation (Fig 3C).

We next measured PHB in several *Comamonas* mutants identified in the screen and found that PHB was undetectable in many mutants when supplemented with lactate (Fig 3D). We did detect a small amount of PHB in some mutants, which is likely insufficient to cause lethality in *C. elegans* although these mutants, like untreated *C. necator,* did induce a developmental delay (Fig 3E). Finally, we fed animals a strain of *C. necator* harboring an early nonsense mutation in the *phaCAB* operon, which has been shown to eliminate PHB production [20]. As expected, this *C. necator* mutant did not produce PHB either in the untreated condition or when supplemented with lactate and it supported full development of *C. elegans* (Fig 3F and 3G).

Thus far, our results uncovered a correlation between bacterial PHB production and lethality in *C. elegans*. To directly test whether PHB production is sufficient to kill *C. elegans*, we engineered *E. coli* HT115 to synthesize PHB by transforming the bacteria with the *phaCAB* operon and the *phaP* phasin gene from *Comomonas*. We confirmed PHB production in the engineered *E. coli* strain by GC-MS (Fig 3H). When supplemented with lactate, the PHB-producing *E. coli* HT115 caused *C. elegans* L1 arrest, lethality, and a Dar phenotype in escapers, to the same extent as *Comamonas* (Fig 3I and 3J). This result demonstrates that bacterial PHB production is sufficient to cause toxicity in *C. elegans*.

## Different PHB-precursors affect *C. elegans* differently depending on bacterial diet

Bacteria can synthesize PHB from a variety of carbon sources [21–23]. We wondered whether toxicity in *C. elegans* in response to different carbon sources was modulated by *Comamonas*, *C. necator*, *E. coli* HT115, or *E. coli*

PLOS Biology

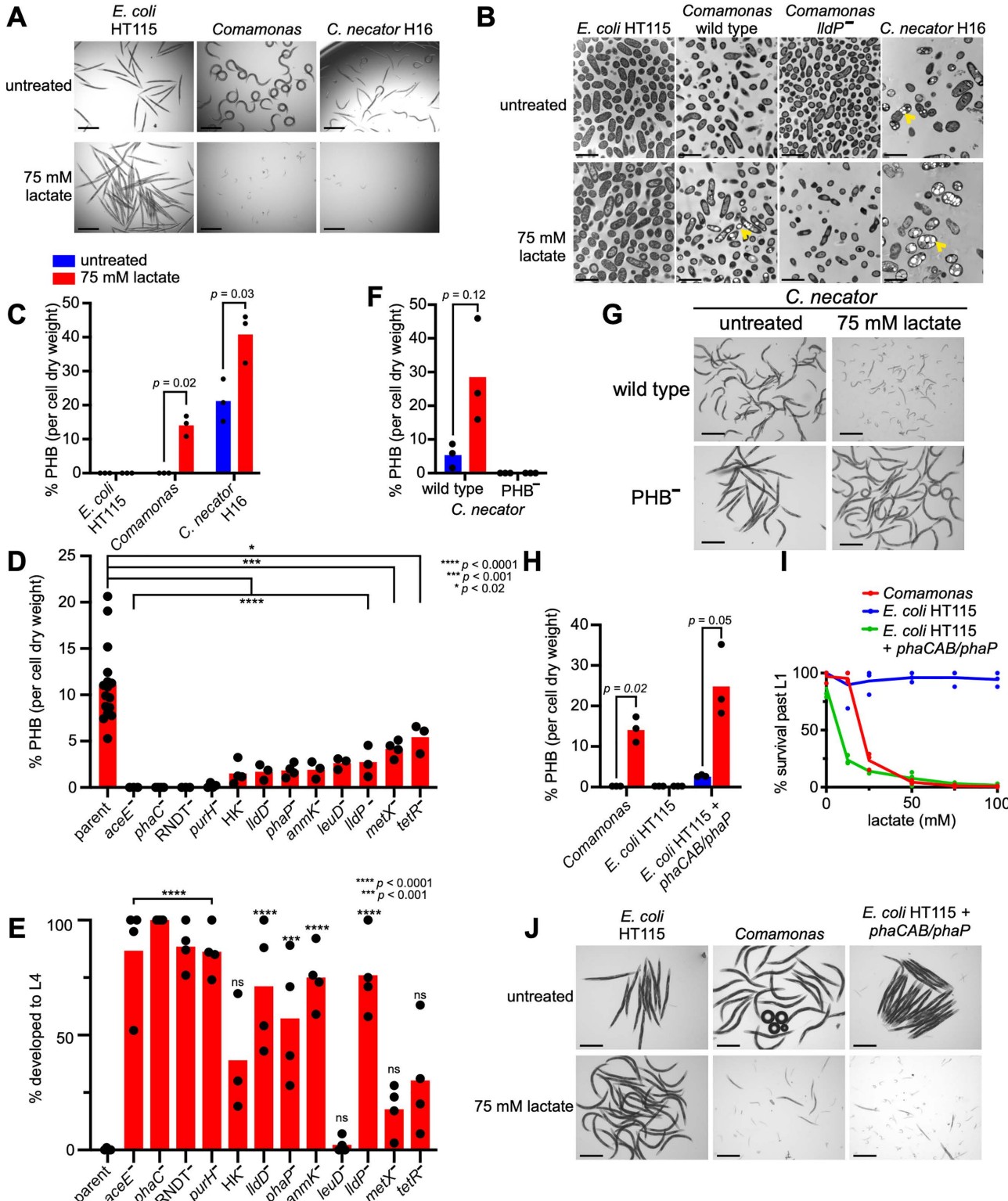

**Fig 3. Bacterial PHB production is necessary and sufficient to kill *C. elegans*. (A)** Brightfield images of *C. elegans* with indicated conditions. Scale bar, 500 μm. **(B)** Transmission electron microscopy (TEM) images of indicated bacteria untreated or supplemented with 75 mM lactate. Arrows indicate PHB granules. Scale bar, 1 μm. **(C)** Gas chromatography-mass spectrometry (GC-MS) measurements of PHB from *E. coli* HT115, *Comamonas,* and

*C. necator* H16 cultured with or without 75 mM lactate. *p* values were determined by unpaired Welch's *t* test. **(D)** GC-MS measurements of PHB from indicated *Comamonas* mutants compared to the parent strain supplemented with 75 mM lactate. *p* values were determined by one-way ANOVA with Dunnett's correction for multiple testing. **(E)** Quantification of development to L4 of animals fed *Comamonas* and selected *Comamonas* mutants supplemented with 75 mM lactate. *p* values for mutants compared to parent determined by one-way ANOVA with Dunnett's correction for multiple testing. **(F)** GC-MS measurements of PHB from *C. necator* wild-type and *phaC* nonsense mutant on untreated and 75 mM lactate conditions. *p* value determined by unpaired Welch's *t* test. **(G)** Brightfield images of *C. elegans* grown on either wild-type or *phaC* mutant *C. necator* plus or minus lactate. Scale bar, 500 µm. **(H)** PHB measurements by GC-MS of *Comamonas*, *E. coli* HT115 and *E. coli* HT115 expressing the *phaCAB* operon and *phaP* gene on untreated and 75 mM lactate conditions. *p* values determined by unpaired Welch's *t* test. **(I)** Dose response curves of lactate toxicity in *C. elegans* fed *Comamonas*, *E. coli* HT115, or *E. coli* HT115 expressing the *phaCAB* operon and *phaP* gene. **(J)** Brightfield images of *C. elegans* with indicated conditions. Scale bar, 500 µm. The underlying data for this figure can be found in S1 Data.

HT115 + *phaCAB* + *phaP*. To test this, we used several different carbon sources known or predicted to be able to produce PHB (Fig 4A). Survival of animals past the L1 stage on sub-lethal doses of metabolites (based on titration data used for Fig 1A) was quantified and compared to the untreated *E. coli* HT115 control. Pyruvate and lactate both caused the greatest lethality for all three PHB-producing bacteria. Interestingly, while animals fed *E. coli* HT115 tolerated all metabolites, consistent with the fact that these bacteria cannot synthesize PHB, the other three bacteria conferred different levels of toxicity in response to different carbon sources. For example, acetate and glucose caused toxicity in animals fed the engineered *E. coli* HT115 strain, whereas gluconate and 3-hydroxypropionate only elicited strong toxicity in animals fed *C. necator*. Interestingly, the engineered *E. coli* HT115 strain differed from the other two PHB-producing bacteria in response to both butyrate and 3-HB; where animals were sensitive to these metabolites when fed *Comamonas* or *C. necator*, they were partially or completely resistant when fed the engineered *E. coli* HT115 strain. These differences in toxicity most likely reflect differences in the uptake and metabolism of these substrates among the different bacteria. For example, unlike *C. necator*, *Comamonas* lacks the GNTK enzyme that converts gluconate to gluconate-6P, a required first step in

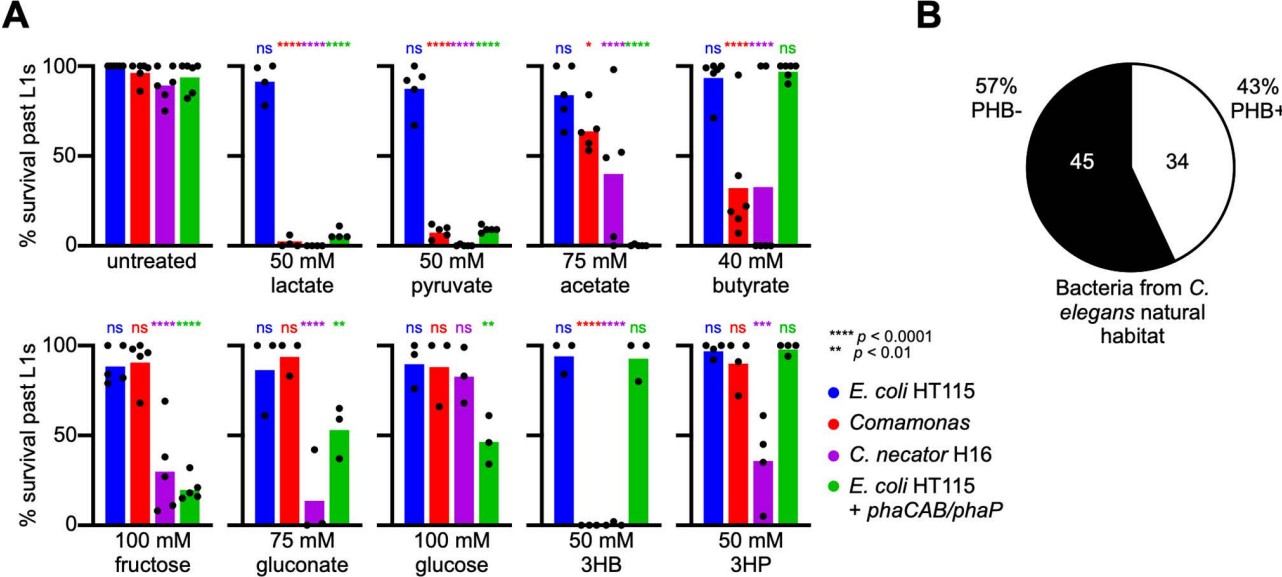

**Fig 4. Bacterial diet modulates the effect of PHB-precursors on *C. elegans*. (A)** Quantification of development past the L1 stage of animals fed indicated bacteria and supplemented with indicated metabolites. *p* values calculated by 2-way ANOVA followed by Dunnett's multiple comparisons tests. Select *p* values comparing each treatment to its untreated control fed the same bacteria denoted by colored stars. ns, not significant; 3-HB, 3-hydroxybutyrate; 3HP, 3-hydroxypropionate. The data underlying this graph can be found in S1 Data. **(B)** Percentage of bacterial species found in *C. elegans* natural habitat that harbor the poly(R)-hydroxyalkanoic acid synthase gene *phaC* required to produce bioplastic PHA and PHB polymers (S3 Table).

gluconate degradation, and consequently cannot utilize it to synthesize PHB [24]. Likewise, *C. necator* cannot metabolize glucose, due to a deficient transport system [20], and indeed this combination does not affect *C. elegans,* while the engineered *E. coli* strain does (Fig 4A).

The composition of the microbial communities that *C. elegans* encounters across its diverse habitat range can affect its population growth and fitness [25–27]. We found that the genomes of several members of *C. elegans'* natural microbiota contain the *phaC* gene, which encodes the poly(R)-hydroxyalkanoic acid synthase, the key enzyme required for the production of PHB polymers and generally used to identify PHB-competent bacteria (Fig 4B and S3 Table) [26,28]. These results show that *C. elegans* likely encounters PHB-producing bacteria in its natural habitat and that these bacteria can produce PHB, depending on which excess carbon source is present in the environment.

### How do PHB-producing bacteria kill *C. elegans*?

We found that the intestine of animals fed PHB-producing bacteria is distended (Figs 1D and S1B). Therefore, we visualized the intestinal lumen using TEM to gain insight into how ingested bacteria producing PHB may block *C. elegans* development and kill the animal. Wild-type animals were imaged after a 6-day exposure to either wild-type *Comamonas* or PHB-deficient *phaC* mutants supplemented with lactate. Because most animals fed PHB-producing *Comamonas* fail to develop beyond the L1 stage, untreated animals were arrested in M9 buffer and imaged as a control (Fig 5A). Animals fed wild-type *Comamonas* supplemented with lactate contained many PHB granules in the intestinal lumen, which was greatly distended (Fig 5A, top). In contrast, animals fed the *Comamonas phaC* mutant were able to develop, their intestinal lumen was not distended and contained food particles and a few undigested bacteria (Fig 5A, middle). Starved L1 animals had empty intestinal lumens (Fig 5A, bottom).

Next, we performed a forward genetic screen in *C. elegans* using the mutagen ethyl methanesulfonate (EMS) to identify mutants that are resistant to PHB-producing bacteria. We screened ~10,000 haploid genomes for mutations that allowed animals to continue developing on *Comamonas* supplemented with lactate. We identified four independent mutants, three of which could not be maintained. The remaining mutant was backcrossed to the wild-type parental strain, and the mutation was identified by whole-genome sequencing. Using the sibling subtraction method [29], we identified a point mutation in the *nuc-1* gene which encodes a homolog of the mammalian DNase II [30] (Fig 5B). To validate the *nuc-1* mutant phenotype, we generated two independent *nuc-1* nonsense alleles by CRISPR-Cas9 genome editing. We found that ~50% of all three *nuc-1* mutants survived to beyond the L4 stage when fed *Comamonas* supplemented with lactate (Fig 5C).

How do mutations in *nuc-1* rescue bacterially produced PHB-induced killing of *C. elegans*? NUC-1 acts in the apoptosis pathway to degrade DNA in dying cells but is also the only nuclease that is secreted into the *C. elegans* gut where it degrades bacterial DNA and RNA [31,32]. Given its function as an excreted, digestive enzyme, we hypothesized that *nuc-1* mutations may affect feeding behavior and/or intestinal functions such as food intake and digestion. We first focused on feeding behavior because the intestinal lumen of animals grown on a PHB-rich diet is enlarged (Figs 1D and S1B) which mimics the distention that occurs when animals are infected by pathogenic bacteria such as *Pseudomonas aeruginosa*. PHB-producing *Comamonas* do increase lawn avoidance, like other pathogenic bacteria [33], however, there was no difference between *nuc-1* mutants and wild-type animals (Fig 5D). *nuc-1* animals food intake is comparable to the wild-type albeit that *nuc-1* mutants had a slightly decreased pumping rate when fed *Comamonas* (Fig 5E and 5F).

We then asked if the number or size of PHB granules was different in *nuc-1* mutants compared to wild-type animals. We exposed wild-type and *nuc-1* mutants to *Comamonas* supplemented with lactate for ~12 hours and imaged them by TEM. Animals were fixed at the mid-late L1 stage and images were taken as cross-sections of the pharynx and anterior intestine. While both strains consumed PHB-containing bacteria, most wild-type animals contained PHB granules in their intestinal lumen, while most *nuc-1* mutant animals did not (Figs 5G and S3A). In addition, we noticed that the pharyngeal

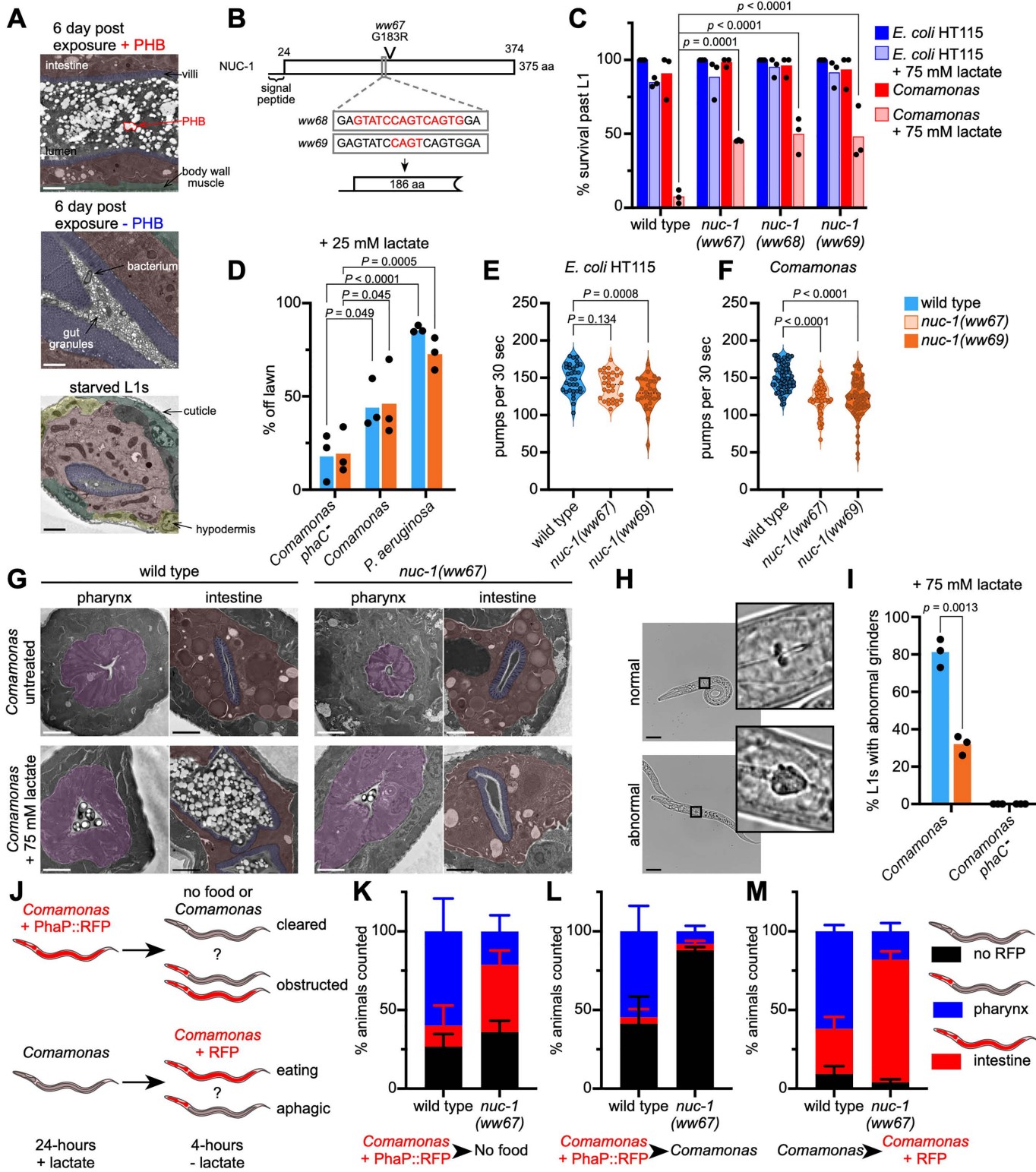

**Fig 5. Bacterially produced PHB obstructs the *C. elegans* digestive tract. (A)** TEM images of *C. elegans* maintained on either wild-type *Comamonas* plus 75 mM lactate or the PHB-deficient *Comamonas phaC* mutant plus 75 mM lactate. Top, arrested L1 animal; Middle, first day adult; Bottom, starved L1 animal. Scale bar, 2 μm. **(B)** Diagram of NUC-1 and mutations created by EMS or CRISPR-Cas9. **(C)** Survival of *nuc-1* mutants compared

to wild-type animals after 48 hours on *E. coli* HT115 and *Comamonas* diets with or without 75 mM lactate. *p* values determined by one-way ANOVA followed by Bonferroni correction for multiple testing. **(D)** Lawn avoidance behavior assay of wild-type and *nuc-1* mutant on *Comamonas* wild-type strain and *phaC* mutant strain grown on 75 mM lactate. *P. aeruginosa* PA14 was used as a positive control. *p* values determined by two-way ANOVA with Dunnett's correction for multiple comparisons. **(E)** Pharyngeal pumping rate of wild-type and *nuc-1* mutants fed *E. coli* HT115. Pooled data from two biological replicates. *p* values determined by one-way ANOVA with Bonferroni correction for multiple testing. **(F)** Pharyngeal pumping rate of wild-type and *nuc-1* mutants fed *Comamonas.* Pooled data from three biological replicates. *p* values determined by one-way ANOVA with Bonferroni correction for multiple testing. **(G)** TEM images of L1 stage wild-type and *nuc-1* mutant animals grown for 12-hours on *Comamonas* supplemented with 75 mM lactate. Scale bar, 2 μm. **(H)** Representative brightfield images of L1 animals with normal and abnormal grinder morphology grown on *Comamonas* with 75 mM lactate. Scale bar, 25 μm. **(I)** Number of wild-type and *nuc-1* mutant animals identified with abnormal pharyngeal grinders. Fifty animals were counted per condition. *P* value was determined by unpaired Welch's *t* test. **(J)** Schematic of diet-switching experiment shown in K, L, and M. **(K–M)** Percentage of animals with RFP observed in pharynx, intestine, or not observed. Fifty animals were counted per condition. The data underlying this figure can be found in S1 Data.

grinders of most (~80%) wild-type L1 animals fed a PHB-rich diet for 24 hours were deformed, while fewer than 40% of *nuc-1* mutant animals showed this phenotype (Fig 5H and 5I).

To test if these effects are associated with changes in food intake, we engineered two strains of *Comamonas* to help visualize the consumption of either bacteria or PHB granules. The first strain expresses the PHB granule-associated protein PhaP fused to RFP ("*Comamonas* + PhaP::RFP") to visualize PHB granules. We used this strain to feed *C. elegans* for 24 hours and then examined whether they were able to clear these bacteria when transferred to either no food or to *Comamonas* that did not contain a fluorescent protein (Fig 5J, top). We found that 60% of wild-type and 21% of *nuc-1* mutant animals retained RFP in the pharynx after being transferred to no food, indicating that PHB granules were stuck in the pharynx and unable to be further digested (Fig 5K). In agreement with this, a greater proportion of *nuc-1* mutant animals had RFP in their intestine than wild-type animals (43% versus 13%). We also transferred animals from *Comamonas* + PhaP::RFP to nonfluorescent *Comamonas* and found that 55% of wild-type animals retained RFP in the pharynx while only 8% of *nuc-1* mutants did. Only a small fraction of animals (4% of both wild-type and *nuc-1* mutant animals) contained PHB in their intestine (Fig 5L).

The second strain expresses RFP ("*Comamonas* + RFP") and was used to directly visualize bacterial uptake in animals that were first fed *Comamonas* supplemented with lactate to see if they could ingest new bacteria (Fig 5J, bottom). We found that 62% of wild-type animals contained RFP exclusively in the pharynx and 29% had RFP throughout the intestine. In contrast, 78% of *nuc-1* mutant animals had RFP throughout the intestine while only 18% had RFP only in the pharynx (Fig 5M). Together, these results show that PHB blocks the flow of food through the pharynx and intestine, and that *nuc-1* mutant animals are better able to digest food after exposure to PHB than wild-type animals.

*nuc-1* mutants accumulate bacterial genomic DNA (gDNA) in the digestive tract [34,35]. We used ethidium bromide (EtBr) staining to confirm the presence of undigested gDNA in the intestinal lumen of *nuc-1(ww67)* mutant animals (S3B Fig). Next, we wondered if an increase in intestinal gDNA could influence the hydrophobicity of PHB, *i.e.,* acting as a laxative. To test this, we tried several in vitro experiments in which we mixed commercially purchased, purified PHB with extracted bacterial gDNA. First, since purified PHB is insoluble in aqueous solvents, we hypothesized that interaction with gDNA might increase its solubility when added to an aqueous PHB suspension. However, no visible change in solubility was observed. We also analyzed the PHB-gDNA mixture by agarose gel electrophoresis stained with EtBr, reasoning that PHB binding to gDNA may either impede or alter gDNA migration on the gel. However, no shift in mobility or reduction of gDNA intensity was detected (S3C Fig).

Unlike purified PHB, the surface layer of PHB-containing carbonosomes within bacteria include the PhaM and PhaR proteins that are known to bind gDNA [36,37]. When consumed by *C. elegans,* PHB may still be associated with these proteins. Therefore, we wondered if, in *nuc-1* mutant animals, these proteins might bind undigested gDNA thereby potentially contributing to the rescue of PHB-induced animal death. Importantly, the *E. coli* HT115 strain engineered to produce PHB only contains the *phaCAB* operon needed to produce PHB and the phasin protein, but lacks the PhaM and PhaR

proteins. If this hypothesis is correct, we would predict that *nuc-1* mutations should fail to rescue animal lethality when fed this engineered strain. However, *nuc-1* mutations did rescue PHB-induced lethality in animals fed this *E. coli* strain, disproving this hypothesis (S3D Fig).

**PHB-producing bacteria compromise intestinal function and permeability**

When *C. elegans* are removed from food, their pumping rate decreases initially, but partly recovers after two hours [38], while their defecation rate slows dramatically [39]. Given the changes in pharyngeal and intestinal morphology, and the difference in the ability to ingest and clear food from the intestine between wild-type and *nuc-1* mutant animals, we next asked whether PHB-containing bacteria may also affect defecation. The *C. elegans* intestine exhibits rhythmic calcium oscillations approximately every 50 s, which drive the DMP (Fig 6A) [40]. This signal can be quantified using an intestinal, calcium-sensing reporter and image analysis [41,42]. We imaged intestinal calcium transients in freely moving animals that were fed *Comamonas* supplemented with or without lactate for 18−24 hours. As expected, wild-type animals fed *Comamonas* without lactate displayed periodic $Ca^{2+}$ waves that propagated rapidly along the length of the intestine (Figs 6B, 6C, S4A, S4B, and S1 Video). The average interval between $Ca^{2+}$ waves of wild-type animals was 53.8 s (Fig 6D and 6E), comparable to animals fed the standard *E. coli* OP50 diet [39], indicating that a *Comamonas* diet alone does not substantially disrupt intestinal calcium oscillations or the DMP. In contrast, animals fed PHB-producing *Comamonas* had dramatic changes to both the period and the propagation of intestinal $Ca^{2+}$ waves (Figs 6B, 6C, S4A, S4B, and S2 Video). While the DMP is typically highly regular, with inter-cycle variability of less than 5 s within individuals, $Ca^{2+}$ oscillations were highly irregular in animals fed PHB-producing bacteria (Fig 6D and 6E). Interestingly, there was a complete absence of $Ca^{2+}$ oscillations in 4 of 24 animals fed PHB-producing bacteria which correlates with a blockage of the digestive tract (Figs 6B, S4A, and S4B). Furthermore, feeding PHB-producing bacteria disrupted the spatial dynamics of $Ca^{2+}$ waves in wild-type animals, causing $Ca^{2+}$ transients to initiate in medial intestinal cells rather than the typical anterior or posterior cells (Figs 6C and S4B). These ectopic $Ca^{2+}$ events failed to propagate to neighboring cells, a phenomenon that is similar to mutants lacking the intestinal gap-junction protein INX-16 [43]. Importantly, wild-type animals fed PHB-deficient *PhaC Comamonas* with or without lactate did not exhibit any changes to the period or propagation of $Ca^{2+}$ waves (Figs 6D, 6E, S4A and S4B). Finally, the $Ca^{2+}$ dynamic changes were rescued by mutations in *nuc-1* (Figs 6B–6E, S4A and S4B, S3 Video and S4 Video).

Given the different intestinal phenotypes we observed, we wondered if PHB may impact intestinal integrity. To test this, we transferred L4 larvae for 24 hours to plates seeded with *Comamonas* supplemented with lactate and then fed the animals blue food coloring, known as a "Smurf Assay" [44]. After consuming the dye, animals were transferred to *E. coli* OP50 for 30 min to flush most of the free dye from the digestive tract. If the intestinal barrier were compromised, the blue color would be retained within the intestinal cells. Most animals fed pore-forming Cry5B-expressing *E. coli* strain, which is known to permeate the intestinal barrier [45], were small and displayed blue coloring within the intestinal cells (Fig 6F and 6G). Strikingly, more than half of animals fed PHB-producing *Comamonas* also contained blue dye in their intestinal cells, while animals fed the PHB-deficient *Comamonas phaC* mutant either showed no dye or color only in the intestinal lumen (Fig 6F and 6G). There was a small but significant difference between *nuc-1* mutant and wild-type animals ($p = 0.022$, Fig 6F and 6G). However, the consistent observation of blue dye in the intestinal cells of *nuc-1* mutant animals fed PHB-producing bacteria indicates that *nuc-1* is not involved in intestinal barrier disruption by PHB. Together, these results indicate that PHB accumulation in the intestine inhibits the DMP, which may compromise the intestinal barrier and contribute to bacterially produced PHB-induced lethality.

## Discussion

Here, we show that bacteria producing the bioplastic PHB kill the nematode *C. elegans* by causing pharyngeal and intestinal impaction. We found several phenotypes associated with killing, including severe changes in pharyngeal grinder

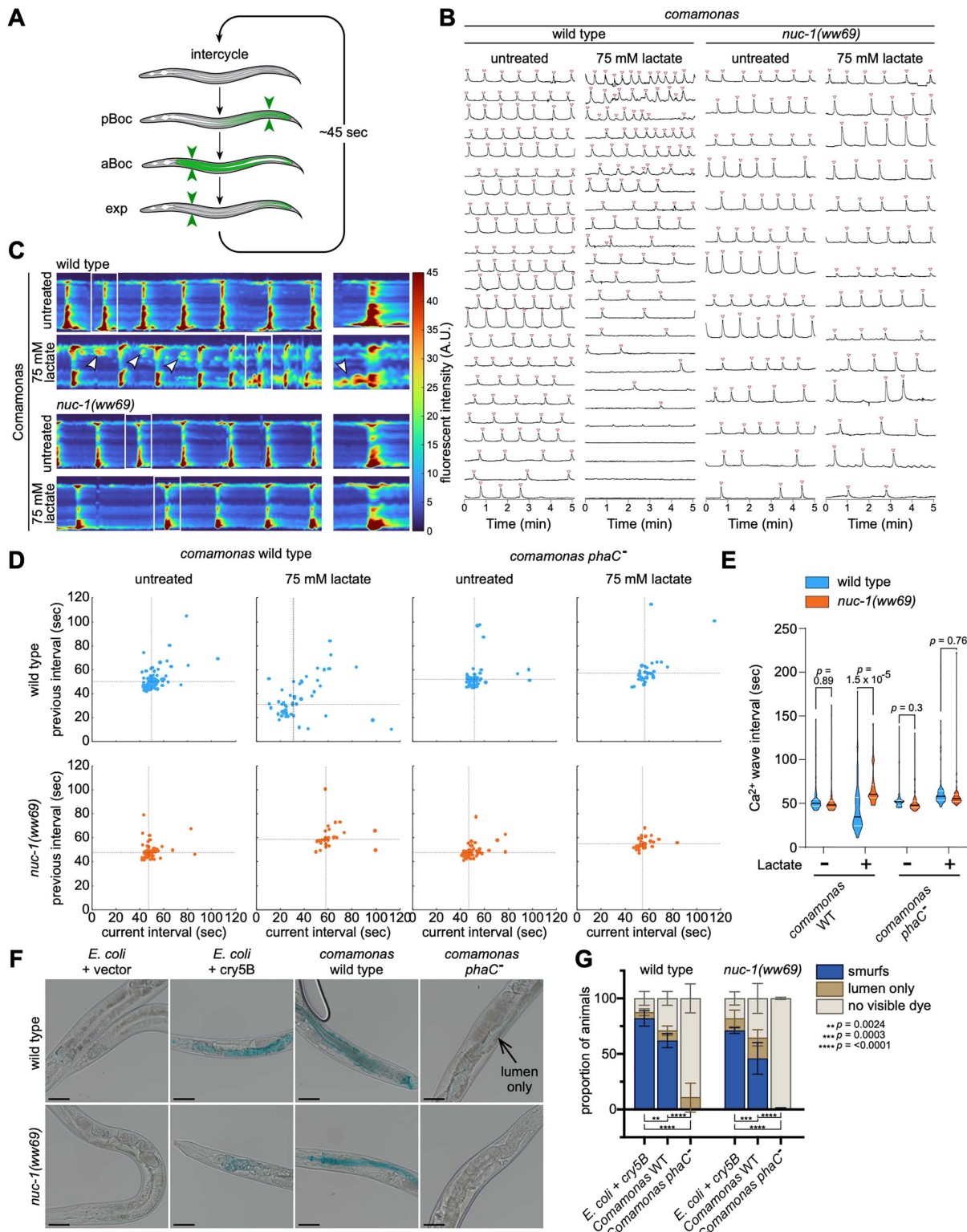

**Fig 6. Acute exposure to PHB-producing bacteria induces intestinal damage in adult animals. (A)** Cartoon of defecation motor program. Green arrows indicate location of muscle contractions. pBoc, posterior body muscle contraction; aBoc, anterior body muscle contraction; exp, expulsion muscle contraction. **(B)** Mean intestinal GCaMP fluorescence in wild-type and *nuc-1* mutants grown for 24-hours on *Comamonas* with or without 75 mM lactate.

Red arrowheads indicate timing of intestinal calcium waves. **(C)** Representative kymographs of intestinal GCaMP fluorescence in wild-type and *nuc-1* mutant animals grown on *Comamonas* wild-type and without 75 mM lactate for 24-hours. Arrows indicate ectopic initiation events. A.U., arbitrary units. **(D)** Comparisons Ca$^{2+}$ wave interval variation of wild-type and *nuc-1* mutant animals grown for 24-hours on either *Comamonas* wild-type or *phaC* mutant with and without 75 mM lactate. **(E)** Quantification of Ca$^{2+}$ wave interval length among wild-type and *nuc-1* mutants grown for 24-hours on *Comamonas* wild-type or *phaC* mutant with and without 75 mM lactate. *P* values were determined by *t* tests with Holm–Šídák correction for multiple comparisons. **(F)** Representative brightfield images of adult wild-type and *nuc-1* mutants treated for 24-hours on indicated condition. Animals were fed *E. coli* expressing Cry5B as a positive control. Scale bar, 50 µm. **(G)** Quantification of number of animals found with blue dye in intestinal tissue, in the lumen exclusively or without dye. Fifty animals were counted per condition. Statistical analysis was performed only on the percentage of fully stained animals using two-way repeated measures ANOVA to evaluate the effects of strain, diet, and their interaction, followed by pairwise comparisons using Bonferroni correction for multiple testing. There was no significant difference found between wild-type and nuc-1 mutants fed either *E. coli* Cry5B ($p = 0.091$) or *Comamonas phaC-* ($p = 0.933$). A small significance was found between wild-type and *nuc-1* mutants fed *Comamonas* wild-type ($p = 0.022$). The underlying data for this figure can be found in S1 Data.

morphology, intestinal barrier dysfunction, and altered defecation (Fig 7). Although physical obstruction may account for the observed phenotypes, it is also possible that PHB alters membrane permeability biochemically. For instance, it has been proposed that PHB aids in the formation and modification of ion channels in bacteria and possibly in eukaryotic membranes [46,47]. Remarkably, about half of the animals were rescued from lethality by mutations in *nuc-1*, and some, but not all PHB-associated phenotypes were alleviated in these animals.

It is not immediately obvious why or how *nuc-1* mutations rescue death induced by ingestion of PHB-producing bacteria. However, given its proposed function as a DNAse that is excreted into the intestinal lumen where it degrades bacterial DNA and RNA, we can speculate about different potential scenarios [31,32]. First, PHB granules can bind DNA in bacteria [48,49] and changes in host bacterial gDNA degradation may affect the amount of DNA bound to PHB in the animal which may aid PHB passage through the digestive tract, like a laxative. We tried to test this hypothesis through several in vitro experiments examining change in solubility of PHB when mixed with bacterial gDNA, or change in gDNA motility through an agarose gel when mixed with PHB. Unfortunately, interpretation of these results was limited by a lack of suitable positive controls. We also supplemented *nuc-1* mutant animals fed PHB-producing bacteria with pre-digested bacterial gDNA, but could not definitively assess any effect on animal survival.

Second, we recently discovered that bacterial RNA is a major energy source for *C. elegans* [50]. It is possible that *nuc-1* mutations affect RNA degradation and/or consumption which may indirectly relate to bacterial PHB-induced death. However, we were not able to confidently assess whether the nuclease activity of NUC-1 in the intestine is required to enhance PHB toxicity.

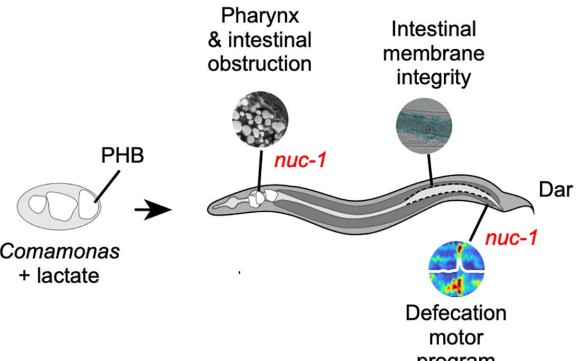

**Fig 7. Bacteria producing PHB harm *C. elegans* in multiple ways.** Model of how bacterially produced PHB affect *C. elegans* and which mechanisms can be alleviated by *nuc-1* knock out. PHB, poly-β-hydroxybutyrate; Dar, deformed anal region.

Our work reveals an unexpected toxic effect of PHB-producing bacteria in an animal. The lethal effects of ingesting bacteria synthesizing PHB on *C. elegans* may have potential implications for ecosystems that depend on symbiotic relationships between bacteria, and between bacteria and animals. *C. elegans* is a bacterivore and many bacteria produce PHB when carbon is in excess to other elements. Nematodes such as *C. elegans* are found in the soil where they feed on decaying vegetation, and they play important roles in nutrient cycling [51–53]. Our study indicates that a large proportion of bacteria found in association with *C. elegans* have the capacity to synthesize PHB. Therefore, bacterially produced PHB may affect *C. elegans* in the wild. However, further studies using different strains of *C. elegans* and other nematodes are needed to determine the potential ecological relevance of our findings.

In this and our previous studies, we have tested only a few bacterial species or strains but already observed dramatic differences in how bacteria influence the physiology of *C. elegans* either as a diet or by modifying different metabolites or compounds. *C. elegans* encounters many bacterial species and conditions in its natural environment. It is likely that a complex network of interactions among bacteria and among bacteria and nutrients shapes host-bacteria interactions. Altogether, *C. elegans* and its bacterial diet provide a simple and effective model to explore how bacteria and nutrients can have different effects in different metabolic contexts.

## Methods

### *Caenorhabditis elegans* and bacterial strains

*Caenorhabditis elegans* were cultured as described [54]. The N2 Bristol strain was used as wild-type. Animals were maintained on Nematode Growth Media (NGM) seeded with *E. coli* HT115 at 20 °C unless otherwise noted. Animals were synchronized for experiments by treating with buffered bleach, followed by L1 arrest overnight at room temperature. Bacterial strains were cultured from a single colony in appropriate antibiotic and media conditions (S1 Table). *C. necator* H16 was purchased from ATCC, and the PHB mutant was purchased from the Leibniz Institute DSMZ collection. *nuc-1* mutant strains VL1536 and VL1537 were made by CRISPR-Cas9 as previously described [55]. Early stop codons were introduced by frameshift-causing indels using the following guide sequence: AAATACGAGTATCCAGTCAG. *nuc-1* CRISPR mutants were backcrossed three times to N2.

### Metabolite supplementation and toxicity calculation

Metabolite stocks were prepared in water, adjusted to pH ~7 and diluted to their final concentration with 1.33X concentrated NGM. Bacterial cultures were seeded at 5X concentration by volume unless otherwise noted. Minimal inhibitory concentrations (MIC) were determined qualitatively by observing developmental delay phenotypes based on body size and morphology relative to animals fed the same diet without metabolite supplementation. Toxicity scores for Fig 1A were calculated by normalizing the MIC to the maximum dose and inverting linearly by subtracting from 1.

### *Caenorhabditis elegans* imaging

Animals were anesthetized with a 1 mM levamisole solution (Sigma) and imaged with an Invitrogen EVOS FL imaging system, a Nikon Eclipse Ci microscope equipped with a Nikon DS-Ri2 camera, a Nikon Eclipse Ti with Andor Zylya camera, or a Leica CTR 5500 microsystem. Images in Figs 1C, 1D, 3B, 5A, 5G, 5H, S1B, and S3B have been adjusted for brightness and contrast using the same parameters for all images per experiment with imageJ.

### PFA-inactivation of bacteria

To inactivate bacterial metabolism, bacteria were fixed with paraformaldehyde (PFA, Fisher) as described previously [56]. Briefly, overnight bacterial cultures were treated with 0.5% PFA for one hour, washed five times with Luria Broth (LB) and

stored as a pellet at −80 °C for up to one month. PFA-treated bacteria pellets were resuspended to an $OD_{600}$ of ~50 in M9 and validated for lack of growth on an LB plate grown overnight at 37 °C.

## Bacterial mutant screen

L1 animals were used to screen a collection of ~5,500 *Comamonas* mutants [11] supplemented with 75 mM lactate (Sigma and BLDpharm, 98% purity) in three biological replicates. Bacterial clones were grown shaking in 96-deep-well plates overnight in LB containing 10 µg/mL gentamycin and 100 µg/mL streptomycin at 37 °C, concentrated 5X by volume in M9 and seeded on 96-well plates containing NGM supplemented with 75 mM lactate. Approximately 25 L1 animals were plated per well and screened visually after 48 and 72 hours. Hits found in all three replicates were streaked for single colonies and the transposon insertion locus was identified by polymerase chain reaction (PCR) using one transposon-specific and one random, degenerative primer (S4 Table), followed by a nested PCR and Sanger sequencing as described [11].

## Electron microscopy

Overnight bacteria cultures were seeded on NGM plates supplemented with or without 75 mM lactate and incubated at room temperature for 24 hours. *C. elegans* were plated on *Comamonas* + 75 mM lactate. Adult animals underwent laser-induced chemical fixation using a MicroPoint ablation system (Andor) on a Nikon Eclipse 80i microscope with a Leica EL6000 light source, a 100x Plan Apo VC lens (1.4 NA) oil objective and Zyla sCMOS camera (Andor). Bacteria and *C. elegans* samples were prepared and imaged by the UMass Chan Medical School Electron Microscopy Core (TEM).

Briefly, samples were fixed in 2.5% glutaraldehyde (v/v) in 0.1 M Sodium cacodylate buffer (pH 7.2) for 60 min at room temperature and then washed three times in fixation buffer and post-fixed for one hour in 1% osmium tetroxide (w/v) in deionized water. Samples were washed with deionized water and then dehydrated in a series of ethanol solutions by 20% increments and a final change of 100% ethanol followed by two changes of propylene oxide. Dehydrated samples were infiltrated overnight in a mixture of infiltrate and 50% propylene oxide/ 50% SPIpon 812 epoxy resin. Following four changes of fresh SPIpon 812 epoxy resin samples were embedded at the ends of beam capsules and polymerized for 48 hours at 68 °C. Using a diamond knife, epoxy blocks were trimmed and then ultrathin sectioned to ~70 nm sections. These sections were then mounted on copper support grids and contrasted with lead citrate and uranyl acetate. Samples were examined on a Philips CM10 transmission electron microscope using 100 kV accelerating voltage. Images were captured using a Gatan TEM CCD camera.

## PHB extraction and quantification

Bacteria were seeded on NGM plates on indicated conditions. After incubation at room temperature for 20 hours, bacteria were harvested in 0.9% sodium chloride (Fisher) and washed three times with water. Lyophilized samples were weighed out in triplicate to around 10 mg and the exact weight was recorded for normalization. PHB was extracted as previously reported with minor adaptations [57,58]. Briefly, PHB was extracted and esterified in 2 mL chloroform (Sigma) and 2 mL of 5% sulfuric acid (Sigma) in methanol containing 0.75 mM methyl benzoate (Sigma) as an internal standard. Reactions were carried out in 16 × 125 mm glass tubes with polytetrafluoroethylene (PTFE) seals incubated at 100 °C for 20 hours. Samples were chilled on ice and aqueous phase separation was performed via the addition of 1 mL of 1 M sodium chloride followed by vortexing for 30 s and then leaving to separate for 20 hours at room temperature. 200 µL of the organic phase was then transferred to GC-MS sampler vials.

Samples were measured on an Agilent 7977B single quadrupole mass spectrometry coupled to an Agilent 7890B gas chromatograph (GC-MS) with a DB-Fatwax Ultra Inert column (30 m × 0.25 mm × 0.25 µm) and helium mobile phase. The injection volume was 0.5 µL with a 5:1 split at an inlet temperature of 250 °C. The oven was set with an initial temperature of 80 °C for 2 min, ramp-up to 245 °C with a rate of 10 °C/min. Methyl-3-hydroxybutyrate was identified based on retention time, 1 quantifier and 2 qualifier ions that were selected manually based on a reference standard. Peak integration was

done with MassHunter software. After blank subtraction, percent PHB was calculated by standard curve and normalized to both the internal methyl benzoate standard and cell dry weight using a custom R script.

All bacteria used to measure were also tested simultaneously on *C. elegans* to confirm the phenotype.

## Molecular cloning and bacterial engineering

The pGG55 construct was created by PCR amplifying the *phaCAB* operon and *phaP* gene from *Comamonas* genomic DNA and inserted stepwise into the backbone vector pZS2 along with T7 promoters and the pMB1 origin from the pUC19 vector by Gibson assembly (S4, S5 Tables). The pDR9 construct was made by inserting the promoter region of rpsL cloned from *Comamonas* upstream of RFP in the pBB1MCS-2 plasmid and the terminator sequence of the *Comamonas* lldPRD operon downstream using Gibson assembly. The pDR14 construct was created by fusing the open reading frame of *phaP1* from *Comamonas* to the N-terminus of RFP in the pDR9 plasmid with the linker sequence GASAGSGSA.

The pGG55 clone was transformed to *E. coli* HT115 and grown in LB with 50 mg/mL kanamycin. All test NGM plates included 50 mg/mL kanamycin to ensure retention of the plasmid and 2 mM IPTG (Fisher) to induce T7 polymerase expression. The pDR9 and pDR14 clones were transformed into *Comamonas* and grown in LB with 100 μg/mL each of Streptomycin and kanamycin.

## *Caenorhabditis elegans* development assays

Approximately 150 synchronized L1 animals were plated in onto indicated conditions. Animals that developed past the L1 stage after roughly 48 hours were quantified manually. Averages were normalized to the average of the *E. coli* HT115 untreated control. All experiments were performed three times, with three technical replicates each.

## *Caenorhabditis elegans* forward genetic screen

The EMS mutagenesis screen was adapted from [59]. Wild-type (N2) animals were exposed to 50 mM ethyl methanesulfonate (EMS, Sigma) for four hours and washed six times with M9 buffer. After recovery on NGM agar plates seeded with *Comamonas*, 200 P0 animals were segregated to new plates. F2 animals were tested for survival on NGM containing *Comamonas* supplemented with 75 mM lactate. Approximately 10,000 haploid genomes were screened, and four homozygous mutants were isolated, one of which remained viable and was backcrossed twice with the parent N2 strain. After identification of the *nuc-1* mutation, this strain was backcrossed four more times to N2 before being used in follow-up experiments.

## Whole genome sequencing and variant mapping

The *nuc-1(ww67)* allele was identified by whole genome sequencing using the sibling subtraction method [29]. Briefly, the *ww67* mutant was crossed to N2 and several homozygous F2s were selected that either exhibited wild-type or mutant phenotype. Genomic DNA was purified by phenol-chloroform extraction and ethanol precipitation. Fragmentation was carried out by a Covaris sonicator M220 and ~300 bp fragments were separated by AMPure beads (Beckman Coulter). Libraries were prepared using the NEBNext Ultra II DNA prep kit (NEB) and sequenced on a NextSeq1000/2000 (Illumina) using a P2 100 cycle kit.

Sibling subtraction analysis was performed using a custom bash script. Briefly, the FASTQ files were aligned to the WS274 *C. elegans* genome using BWA-MEM [60]. Resulting BAM files were sorted and processed using SAMtools, followed by variant detecting using BCFtools [61]. Variants unique to the mutant samples were filtered for using BEDTools [62]. Variants were annotated and predicted effects reported using snpEFF [63]. The mutation was confirmed by Sanger sequencing.

## Gut staining with EtBr

In vivo gut DNA staining was adapted from [34]. Mixed stage animals were washed off plates into microfuge tubes and then washed until clear. Animals were resuspended in 100 μL of 10 X *Comamonas* plus 1 μL of EtBr (10 mg/mL) and

incubated for 30 min before being washed with phosphate-buffered saline. Animals were anesthetized using levamisole and imaged using phase contrast and the TX red filter on a Nikon Ti microscope with an Andor Zyla camera and Sola SE 5-LCR-SB light source.

## Behavioral assays

Lawn avoidance assays were conducted by picking 50 L4 animals grown on untreated plates seeded with *Comamonas* wild-type or the *phaC* mutant to untreated, unseeded plates for 30-min and then picked to treated plates with equivalent bacteria spotted to the center of the plate. Animals were counted ~24-hours later.

For pumping rate, ~20 first-day adults grown on *Comamonas* or *E. coli* HT115 were singled to a plate with a thin-lawn of the corresponding bacteria and allowed to acclimate for at least 1 hour. Around 15 individual animals were videoed using a CMOS camera (BFLY-U3-23S6M-C) with an Olympus XLFluor 4x 240 (0.28 NA) objective for at least 30 s at 20 frames per second (FPS). Pumping rate was counted at 10 FPS for 600 frames total (30 s).

## Intestinal calcium imaging

Wild-type and *nuc-1* mutant animals expressing the intestinal GCaMP reporter *zfIs178[Pges-1::NLSwCherry::SL2::GCaMP6]* were raised on *E. coli* OP50. L4 animals were transferred to *Comamonas* with or without 75 mM lactate the day prior to recording. Since thick bacterial lawns interfered with downstream $Ca^{2+}$ imaging analysis, thin-lawn plates were prepared by scraping the bacteria off a culture plate and diluting them 1:1 with M9 buffer and spotting 100 μl of the mixture on a 6-cm plate and allowing it to dry at room temperature. Young adult animals were transferred to thin-lawn plates seeded with the corresponding bacteria and allowed to acclimate for at least 10 min prior to imaging. Two CMOS cameras (BFLY-U3-23S6M-C, Teledyne-FLIR) were used to acquire simultaneous brightfield and fluorescent images with an Olympus XLFluor 4x 340 (0.28 NA) objective. Brightfield illumination was provided by a 760 nm infrared led (LED760L, Thorlabs) in conjunction with a 685 nm filter (FF685-Di02-25x36, Semrock). Illumination for GCaMP imaging was provided by a 470 nm LED (M470L5, Thorlabs) using a 470/28 nm excitation filter (FF01-470/28-25, Semrock), a 495 nm dichroic filter (FF495-Di03-25x36, Semrock) and a 525/40 nm emission filter (FF02–525/40-25, Semrock). Images were acquired at 15 FPS with a 66 ms exposure time and 2×2 image binning for a 5-min period.

Intestinal GCaMP dynamics were measured using custom Matlab scripts (Mathworks). The bulk $Ca^{2+}$ signal was obtained by thresholding the brightfield image using Otsu's method [64] to obtain a binary mask of the body of the worm. This mask was refined through morphological dilation and erosion and used to measure the mean gray values of the corresponding pixels in the GCaMP image. The mean value of pixels outside the binary mask was subtracted from the mean value within the mask to yield a background subtracted bulk $Ca^{2+}$ signal. The Matlab 'findpeaks' function was used to identify spikes in the bulk signal to determine intervals between $Ca^{2+}$ waves and verified by manual inspection of bulk signal and kymograph data.

Kymographs were generated by skeletonization of the binary mask to yield a midline through the length of the animal. This midline was divided into 100 segments and normal vectors were calculated for the midpoint of each segment. The pixel intensity along each of these perpendicular lines was measured, yielding a straightened animal. The maximum value at each of these 100 segments was recorded and resampled to 200 points using linear interpolation for better visualization of kymographs. The anterior end of the animal was determined based on the absence of GCaMP expression in the pharynx and the orientation of the kymograph was corrected based on this feature, however accurate kymographs could not be produced during frames when the animal was in a coiled posture.

## Smurf assay

Around 500 L4 animals fed either wild-type *Comamonas*, the *phaC* mutant or *E. coli* BL21(DE3) vector (control for *E. coli* expressing Cry5B) were washed six times with M9 buffer and then transferred to plates containing 75 mM lactate seeded either with *Comamonas* wild-type or *phaC* mutant or transferred to plates containing 1 mM IPTG seeded with the positive

control, *E. coli* expressing Cry5B, a pore-forming protein known to cause defects in intestinal membrane integrity [45]. Animals and most of their bacterial lawn were harvested in 400 µL of M9 ~24-hours later and 100 µL of 5% FD&C Blue dye #1 (Sunnest) was added to a final concentration of 1%. Samples were covered and rocked for three hours before being washed twice with M9 and plated to untreated plates seeded with *E. coli* OP50 and allowed to recover for 30-min to 1-hour. Live animals were picked to M9 buffer containing 1 mM levamisole (Sigma) and mounted on 2% agarose slides. Animals were imaged on a Nikon Eclipse Ci microscope equipped with a Nikon DS-Ri2 color camera using a 40X objective. Animals were scored for presence of blue food coloring in the intestinal cells or in the intestinal lumen.

## Statistical analysis

All experiments were performed with at least three biological replicates except Fig 4F, which was carried out twice, and S2C Fig, which was done once. Statistics were calculated using GraphPad Prism version 9.5.

## Supporting information

**S1 Data. Tabulated raw data.** Raw data underlying Figs 1A, 2D, 3C–3F, 3H, 3I, 4A, 5C–5F, 5I, 5K–5M, 6D, 6E, S1A, S2B, S2D, S3D.
(XLSX)

**S1 Table. Bacteria strain information.** Bacteria species and strains used in this study, along with growth conditions and origins. Related to Fig 1.
(XLSX)

**S2 Table. *Comamonas* mutant hits.** List of mutant hits from *Comamonas* genetic screen related to Fig 2.
(XLSX)

**S3 Table. *Caenorhabditis elegans* microbiota.** List of bacteria species isolated alongside *C. elegans*. Related to Fig 3.
(XLSX)

**S4 Table. Oligos used in this study.**
(XLSX)

**S5 Table. Plasmids used in this study.**
(XLSX)

**S1 Video. Calcium sensor in wild-type *Caenorhabditis elegans* fed *Comamonas*.**
(MP4)

**S2 Video. Calcium sensor in wild-type *Caenorhabditis elegans* fed PHB-produicing *Comamonas*.**
(MP4)

**S3 Video. Calcium sensor in *nuc-1* mutant animals fed *Comamonas*.**
(MP4)

**S4 Video. Calcium sensor in *nuc-1* mutant animals fed PHB-producing *Comamonas*.**
(MP4)

**S1 Fig. Different bacteria diets modulate metabolite toxicity.** (**A**) MIC was determined qualitatively upon first observance of small or slow developing animals in the population. The data for this table can be found in S1 Data. (**B**) Brightfield images of adult animals. Orange dashed lines indicate intestinal lumen. Scale bar, 50 µm.
(TIFF)

**S2 Fig. Bacterial PHB levels correlate with phenotype severity in *Caenorhabditis elegans*.** (**A**) Quantification of animal survival past the L1 stage. *p* values determined by unpaired Welch's *t* test. (**B**) Quantification of animal development to the L4 stage. *p* values determined by unpaired Welch's *t* test. (**C**) Brightfield images of adult animals grown on *Comamonas* plus/minus lactate. Scale bar, 100 μm. (**D**) GC-MS measurements of PHB from five different *C. necator* H16 colonies grown with or without 75 mM lactate. Brightfield images of animals grown on bacteria from the same *C. necator* colonies without lactate are shown below the bar graph. Scale bar, 500 μm. The data underlying this figure can be found in S1 Data.
(TIFF)

**S3 Fig. Bacterially produced PHB granules accumulate in *Caenorhabditis elegans*.** (**A**) Wild-type and *nuc-1* mutant animals grown on *Comamonas* supplemented with lactate for 12 hours and imaged as cross-sections by TEM. Scale bar, 2 μm. (**B**) Phase and fluorescent images of mixed stage animals grown on *Comamonas* and stained with EtBr. Intensity of fluorescence seen along intestinal lumen indicated undigested gDNA. Scale bar, 100 μm. (**C**) One percent agarose gel with EtBr loaded with 50 μL PHB-gDNA mixture per well, where indicated. (**D**) Survival rate of wild-type and *nuc-1* mutants grown on *Comamonas*, *E. coli* HT115, or *E. coli* HT115 + *phaCAB/phaP* with or without lactate. *P* values determined by two-tailed Welch's *t* test. The underlying data for this graph can be found in S1 Data.
(TIFF)

**S4 Fig. A PHB-rich diet disrupts Ca$^{2+}$ waves in the *Caenorhabditis elegans* intestine.** (**A**) Traces measuring mean intestinal GCaMP fluorescence in wild-type and *nuc-1* mutants grown for 24-hours on *Comamonas* and the *phaC* mutant with and without 75 mM lactate. (**B**) Kymographs of intestinal GCaMP fluorescence from all animals measured of wild-type and *nuc-1* mutants grown on *Comamonas* and the *phaC* mutant with and without 75 mM lactate for 24-hours. A.U., arbitrary units. The data underlying this figure can be found in S1 Data.
(TIFF)

**S1 Raw Image. Raw image of agarose gel stained with EtBr used in S3 FigC.**
(JPG)

## Author contributions

**Conceptualization:** Gabrielle E. Giese, Albertha J. M. Walhout.

**Data curation:** Gabrielle E. Giese, Daniel M. Richards.

**Formal analysis:** Gabrielle E. Giese, Daniel M. Richards, Jeremy T. Florman, Daniel J. Durning.

**Funding acquisition:** Mark J. Alkema, Albertha J. M. Walhout.

**Investigation:** Gabrielle E. Giese, Daniel M. Richards, Jeremy T. Florman, Alyxandra N. Starbard, Ann A. Xu.

**Methodology:** Gabrielle E. Giese, Jeremy T. Florman.

**Project administration:** Albertha J. M. Walhout.

**Supervision:** Mark J. Alkema, Albertha J. M. Walhout.

**Validation:** Gabrielle E. Giese, Daniel M. Richards, Alyxandra N. Starbard, Ann A. Xu.

**Writing – original draft:** Gabrielle E. Giese, Daniel M. Richards, Albertha J. M. Walhout.

**Writing – review & editing:** Gabrielle E. Giese, Daniel M. Richards, Jeremy T. Florman, Ann A. Xu, Mark J. Alkema, Albertha J. M. Walhout.

## Acknowledgments

We thank members of the Walhout lab, Job Dekker, Amy Walker, and Caryn Navarro for discussion and critical reading of the manuscript. We also thank members of the Brewster lab for microscopy help and sharing the pZS2 plasmid. We also thank the Aroian lab for sharing the *Escherichia coli* BL21(DE3) vector and Cry5B strains, Amy Walker and Victor Ambros for bacterial strains, and Leonard Barassa from the Thompson lab for technical expertise.

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
