## [Editor Report · Decision Letter 0]

5 Feb 2026

Dear Dr Walhout,

Thank you for submitting your manuscript entitled “Bioplastic producing bacteria kill the nematode C. elegans” for consideration as a Research Article by PLOS Biology under our Portable Peer Review process.

Your manuscript has now been evaluated by the PLOS Biology editorial staff and discussed with an Academic Editor with relevant expertise. Based on this initial assessment, we would be interested in inviting you to proceed further with your submission. However, I would like to clarify how we would proceed should you wish to continue.

At this stage, no changes to the manuscript are required for submission. We ask that you complete your submission by providing the required metadata only. Once this step is completed, we would issue a Major Revision decision, outlining a set of substantial textual and interpretive revisions that we consider necessary for the manuscript to be suitable for publication at PLOS Biology. The revised manuscript would then be assessed by the Academic Editor after the revision period, without a further round of external peer review.

In particular, while the study presents compelling evidence that bacterial production of poly-β-hydroxybutyrate (PHB) is necessary and sufficient to induce lethality and physiological disruption in C. elegans, we have concerns about how this finding is currently framed and interpreted throughout the manuscript. As written, the title, abstract, Results section headings, and parts of the Discussion give the impression that the study addresses the toxicity of “bioplastics” or PHB itself. However, the data support a more specific conclusion: that C. elegans dies when fed bacteria that produce PHB (rather than PHB by itself), under the experimental conditions tested. We consider this to be an important finding in its own right, without the need to overstate the implications.

The forthcoming Major Revision decision would therefore request revisions that more closely align the claims with the data. This would include, for example, revising the title to focus on PHB-producing bacteria rather than bioplastics more broadly, and adjusting the abstract and Results language to remove or modify statements that imply direct toxicity of PHB or bioplastics per se, among other related changes.

We recognize that similar concerns were raised during prior peer review at Cell, and we appreciate the additional experiments and clarifications you have already provided. While we do not anticipate requesting further experimental work, we would require these textual and interpretive changes for the manuscript to be suitable for publication at PLOS Biology.

If you are willing to proceed under these conditions, we would be happy to invite completion of the submission. If you feel that these changes would not be compatible with your intended presentation of the work, you may of course choose not to proceed. Should you decide to continue, we would also appreciate it if you could provide the formal rejection letter from Cell, in addition to the response to the reviewers.

Once your full submission is complete, your paper will undergo a series of checks. To provide the metadata for your submission, please Login to Editorial Manager (https://www.editorialmanager.com/pbiology) within two working days, i.e. by Feb 07 2026 11:59PM.

Kind regards,

Melissa

Melissa Vazquez Hernandez, Ph.D.

Associate Editor

PLOS Biology

---

## [Editor Report · Decision Letter 1]

11 Feb 2026

Dear Marian,

Thank you for your patience while your manuscript "Bioplastic producing bacteria kill the nematode C. elegans" was peer-reviewed at PLOS Biology.

As I mentioned before, your manuscript has now been evaluated as Portable Peer Review from Cell by the editorial team in consultation with the Academic Editor. We find the experimental work strong and the evidence convincing that bacterial production of poly-β-hydroxybutyrate (PHB) is necessary and sufficient to induce lethality and physiological disruption in C. elegans.

However, as I mentioned in the invitation letter, we have concerns about how the findings are framed throughout the manuscript. As written, the title, abstract, Results headings, and parts of the Discussion give the impression that the study demonstrates toxicity of “bioplastics” or PHB itself. In contrast, the data support a more specific conclusion: that C. elegans dies when fed bacteria that produce PHB under the experimental conditions tested.

While no additional experiments are required; we do require substantial textual revisions to align the claims with the data.

In particular, we request:

- revision of the title to focus on PHB-producing bacteria rather than “bioplastics” broadly.

- removal or revision of the Abstract statement (lines 23–24) referring to “non-toxic” biodegradable bioplastics.

- systematic adjustment of language throughout the Results and Discussion (e.g., line 170; lines 510–512) to avoid implying direct toxicity of purified PHB or commercial bioplastics, unless it is experimentally proven

- clear framing of broader environmental or biomedical implications as speculative rather than demonstrated

More specific *examples* and comments from the Academic Editor can be found at the end of this e-mail.

Given the extent of revision needed, we cannot make a decision about publication until we have seen the revised manuscript and your response to the reviewers' comments. Your revised manuscript is likely to be sent for further evaluation by all or a subset of the reviewers.

**IMPORTANT - SUBMITTING YOUR REVISION**

*Re-submission Checklist*

*Published Peer Review*

*PLOS Data Policy*

*Blot and Gel Data Policy*

Sincerely,

Melissa

Melissa Vazquez Hernandez, Ph.D.

Associate Editor

PLOS Biology

From the Academic Editor:

Title: To me, the most important thing to change would be the title. I'd be OK with "bacterial production of poly-β-hydroxybutyrate kills the nematode C. elegans" or "poly-β-hydroxybutyrate-producing bacteria kill the nematode C. elegans" or along the lines what you had suggested as well.

Abstract: In the abstract, ideally, the sentence on bioplastics would be removed ("One such polymer, poly-β-hydroxybutyrate (PHB), is used commercially to produce ‘non-toxic’, biodegradable bioplastics."). At bare a minimum, to avoid the appearance that this study is about bioplastic toxicity, the word "non-toxic" should be removed.

Throughout the results: Replace "bioplastics", "bioplastic-producing bacteria", etc. with the specific compound name, i.e. "PHB" and "PHB-producing bacteria", etc.

→ Other examples that can be used to consider the modifications:

Chaning names of subsections in the Results and Figures similar to "PHB kills C-elegans" for something similar to "C. elegans dies when bacteria it produces PHB"

---

## [Editor Report · Decision Letter 2]

5 Mar 2026

Dear Marian,

Thank you for your patience while we considered your revised manuscript "Bacteria producing the bioplastic polyhydroxybutyrate kill the nematode C. elegans" for publication as a Research Article at PLOS Biology. This revised version of your manuscript has been evaluated by the PLOS Biology editors, and the Academic Editor.

Based on our Academic Editor's assessment of your revision, we are likely to accept this manuscript for publication. Please make sure to address the following data and other policy-related requests.

1) Thank you for the change in the title. We had further discussion within the team and we would like to suggest the following title to bring up also the host-microbe angle:

"Bacteria producing the biodegradable bioplastic polyhydroxybutyrate can kill their host nematode C. elegans"

2) Please add the weblink of the funding agencies in the Financial Disclosure statement in the manuscript details.

3) Please add to your Competing Interests the following statement “MA is a member of PLOS Biology’s Editorial Board. The other authors declare that no competing interests exist."

Please supply the numerical values either in the a supplementary file or as a permanent DOI’d deposition for the following figures:

Figure 1A, 2DE, 3CFDHIE, 4A, 5CDEFIKLM, 6DEG, S1A, S2ABD, S3D

5) Please cite the location of the data clearly in all relevant main and supplementary Figure legends, e.g. “The data underlying this Figure can be found in S1 Data” or “The data underlying this Figure can be found in https://doi.org/10.5281/zenodo.XXXXX”

6) Supplementary files (e.g., excel). Please ensure that all data files are uploaded as 'Supporting Information' and are invariably referred to (in the manuscript, figure legends, and the Description field when uploading your files) using the following format verbatim: S1 Data, S2 Data, etc. Multiple panels of a single or even several figures can be included as multiple sheets in one excel file that is saved using exactly the following convention: S1_Data.xlsx (using an underscore).

7) Please ensure that your Data Statement in the submission system accurately describes where your data can be found and is in final format, as it will be published as written there

8) Per journal policy, if you have generated any custom code during the course of this investigation, please make it available without restrictions. Please ensure that the code is sufficiently well documented and reusable, and that your Data Statement in the Editorial Manager submission system accurately describes where your code can be found. More information on our Code Policy, what and how to share can be found here: https://journals.plos.org/plosbiology/s/code-availability

***Insert bulleted list of Editorial Requirements HERE, including REPORTING, ETHICS & DATA REQUIREMENTS AS NECESSARY (BRING THEM UP FROM BELOW THE SIGNATURE, AS THEY ARE MUCH LESS LIKELY TO BE IGNORED)***

We expect to receive your revised manuscript within two weeks.

*Published Peer Review History*

*Press*

Sincerely,

Melissa

Melissa Vazquez Hernandez, Ph.D.

Associate Editor

PLOS Biology

---

## [Editor Report · Decision Letter 3]

24 Mar 2026

Dear Dr Walhout,

Thank you for the submission of your revised Research Article "Bacteria producing the bioplastic polyhydroxybutyrate kill the nematode C. elegans" for publication in PLOS Biology. On behalf of my colleagues and the Academic Editor, Sebastian Winter, I'm pleased to say that we can in principle accept your manuscript for publication, provided you address any remaining formatting and reporting issues. These will be detailed in an email you should receive within 2-3 business days from our colleagues in the journal operations team; no action is required from you until then. Please note that we will not be able to formally accept your manuscript and schedule it for publication until you have completed any requested changes.

Sincerely,

Roli Roberts

Roland G Roberts, PhD

Senior Editor

PLOS Biology

rroberts@plos.org

on behalf of

Melissa Vazquez Hernandez, Ph.D., Ph.D.

Associate Editor

PLOS Biology
